# NPC Intracellular Cholesterol Transporter 1 Regulates Ovarian Maturation and Molting in Female *Macrobrachium nipponense*

**DOI:** 10.3390/ijms25116049

**Published:** 2024-05-31

**Authors:** Sufei Jiang, Wenyi Zhang, Yiwei Xiong, Mengying Zhang, Huwei Yuan, Yunpeng Niu, Hui Qiao, Hongtuo Fu

**Affiliations:** 1Key Laboratory of Freshwater Fisheries and Germplasm Resources Utilization, Ministry of Agriculture and Rural Affairs, Freshwater Fisheries Research Center, Chinese Academy of Fishery Sciences, Wuxi 214081, China; jiangsf@ffrc.cn (S.J.); zhangwy@ffrc.cn (W.Z.); xiongyw@ffrc.cn (Y.X.); 2Wuxi Fisheries College, Nanjing Agricultural University, Wuxi 214081, China; zmy20000310@163.com (M.Z.); yuan08102021@126.com (H.Y.); niuyp5259@163.com (Y.N.)

**Keywords:** *Macrobrachium nipponense*, NPC intracellular cholesterol transporter 1, ovarian maturation

## Abstract

NPC intracellular cholesterol transporter 1 (NPC1) plays an important role in sterol metabolism and transport processes and has been studied in many vertebrates and some insects, but rarely in crustaceans. In this study, we characterized NPC1 from *Macrobrachium nipponense* (*Mn-NPC1*) and evaluated its functions. Its total cDNA length was 4283 bp, encoding for 1344 amino acids. It contained three conserved domains typical of the NPC family (NPC1_N, SSD, and PTC). In contrast to its role in insects, *Mn-NPC1* was mainly expressed in the adult female hepatopancreas, with moderate expression in the ovary and heart. No expression was found in the embryo (stages CS–ZS) and only weak expression in the larval stages from hatching to the post-larval stage (L1–PL15). *Mn-NPC1* expression was positively correlated with ovarian maturation. In situ hybridization showed that it was mainly located in the cytoplasmic membrane and nucleus of oocytes. A 25-day RNA interference experiment was employed to illustrate the Mn-NPC1 function in ovary maturation. Experimental knockdown of *Mn-NPC1* using dsRNA resulted in a marked reduction in the gonadosomatic index and ecdysone content of *M. nipponense* females. The experimental group showed a significant delay in ovarian maturation and a reduction in the frequency of molting. These results expand our understanding of NPC1 in crustaceans and of the regulatory mechanism of ovarian maturation in *M. nipponense*.

## 1. Introduction

*Macrobrachium nipponense* (commonly known as the oriental river prawn) is widely distributed in Southeast Asia [1] including China. It is very popular in Chinese cuisine due to its delicious taste and high nutritional value [2]. It is an important commercial freshwater crustacean in Chinese aquaculture and plays an increasingly widespread role in the revitalization of rural economies. *Macrobrachium nipponense* has a short sexual maturation period, enabling “autumn reproduction” during the breeding season (especially in late summer). The newly hatched prawns only need about 45 days to grow to sexual maturity and be ready for “autumn reproduction”. Multi-generation populations therefore develop in rearing ponds over a short period of time, creating an increased food demand, increased risk of hypoxia, and reduction in individual prawn size. “Autumn reproduction” has a major impact on the economic benefits of prawn culture and increases farmers’ incomes [3]. To address this issue, previous studies have attempted to explore the potential mechanisms governing rapid ovarian development in *M. nipponense*. Many ovarian expressed sequence tags (ESTs) and transcriptome libraries have been constructed [1,3] in screening studies for many candidate genes closely related to ovarian development, such as vitellogenin, gonad-inhibiting hormone, cathepsin L, vitellogenin receptor, cyclin A, ribosomal protein L24, cyclooxygenase nucleoside, and diphosphate kinase [4,5,6,7]. However, the regulatory mechanisms governing rapid ovarian maturation in *M. nipponense* remain poorly understood.

The hepatopancreas plays an important role in crustacean ovarian maturation by providing energy, essential fatty acids, and cholesterol for steroid hormone synthesis [8]. The hepatopancreas is also the main site of vitellogenin synthesis in *M. nipponense*, as well as in *Scylla paramamosain*, *Penaeus monodon*, *Eriocheir sinensis*, and *Carcinus maenas* [9,10,11,12]. We performed a comparative hepatopancreatic proteomic analysis of female *M. nipponense* during five ovarian maturation stages to explore the differentially expressed proteins and pathways regulating ovarian maturation [13]. NPC intracellular cholesterol transporter 1 (NPC1) was found to be the top up-regulated protein in the vitellogenesis stage (ovary stages II to III) and the most down-regulated protein during the emptying stage (ovary stages IV to V), indicating its significant role in regulating ovarian maturation.

NPC1 is conserved and widely distributed in both vertebrates and invertebrates [14]. In vertebrates, low-density lipoprotein particles are taken up by cells and sent to lysosomes for decomposition by acid lipase [15,16]. The cholesterol released is exported by transporters (NPC1 and NCP2) to cells via the lysosomes [17,18]. In vertebrates, NPC1 is highly expressed in the intestinal epithelial cells. An inherited autosomal disease results in mutations of NPC1 and causes significant accumulation of cholesterol and glycosphingolipids [19]. The NPC1 protein has only been rarely reported in insects. Two vertebrate homologs (NPC1a and NPC1b) have been found in *Drosophila* spp. NPC1a is widely distributed in the tissues whereas NPC1b is only expressed in the midgut epithelial cells. NPC1a is responsible for intracellular sterol transport, while NPC1b is involved in dietary sterol absorption [20,21]. NPC1 has also been shown to be involved in spermatogenesis in *Drosophila* spp. Specifically, male sterility was completely rescued using transgenic techniques using a widely expressed promoter to drive NPC1 gene expression in the NPC1 mutant [22]. In *Bombyx mori*, the expression of BmNPC1 was greatest in the testis, malpighian tubules, hemocytes, and ovaries. Knockdown of BmNPC1 caused cholesterol to accumulate in the cells and also prevented molting in the larvae [23]. In *Bemisia tabaci*, BtNPC1 was widely expressed at all developmental stages and the highest level was found in the adult abdomen. BtNPC1 knockdown led to reduced survival and fecundity. Knockdown of BtNPC1 also affected the development and metamorphosis of *B. tabaci* nymphs [24]. However, NPC1 has not yet been studied in crustaceans.

In this study, we analyzed the coding cDNA sequence structure and the evolutionary development of the amino acid sequence of NPC1 in *M. nipponense* (*Mn-NPC1*). We analyzed its expression profiles and subcellular localization in various tissues and at various developmental stages. We also investigated its function in regulating ovarian maturation using RNA interference (RNAi) technology, to build a better theoretical reference for exploring the mechanism of ovarian maturation in *M. nipponense*.

## 2. Results

### 2.1. Sequence Analysis of Mn-NPC1

The open reading frame (ORF) of *Mn-NPC1* was validated using PCR sequencing. Its total cDNA length was 4283 bp, with 5′ and 3′ untranslated regions of 60 and 188 bp, respectively, and a Genbank No. of PP150339. The ORF of *Mn-NPC1* was 4035 bp and coded for 1344 amino acids (Appendix A). The theoretical pI of *Mn-NPC1* was 4.89 and the predicted molecular mass was 148.33 KDa. Its precursor peptide consisted of a 26 amino acid signal peptide and 1318 amino acid mature peptide, containing a total of 16 N-glycosylation sites and 12 typical transmembrane domains. An analysis of the *Mn-NPC1* conserved domains revealed that it had three typical NPC family conserved domains: an NPC1 N-terminal domain (NPC1_N, at 34–282 aa), a sterol-sensing domain (SSD, at 658–812 aa), and a patched domain (at 1005–1266 aa).

### 2.2. Sequence Alignment and Phylogenetic Analysis

A sequence alignment analysis of the NPC1 amino acid sequences of crustaceans, insects, and higher vertebrates was performed using DNAMAN 6.0 (https://dnaman.software.informer.com/6.0/, accessed on 12 April 2023, Appendix A). The results showed that the three NPC family typical conserved domains were highly conserved in all of the species studied. The shared amino acid identities of Mn-NPC1 with *Procambarus clarkii*, *Cherax quadricarinatus*, *P. monodon*, *Litopenaeus vannamei*, *P. chinensis*, and *E. sinensis* were 54.69%, 55.10%, 53.29%, 53.78%, 53.15%, and 51.12%, respectively. The shared sequence identities of more than 50% indicated that NPC1 is highly conserved in crustaceans. The shared Mn-NPC1 amino acid identities with the model insect’s *D. melanogaster* and *B. mori* were 39.72% and 39.37%, respectively. The amino acid homology of Mn-NPC1 with Homo sapiens and Danio rerio were 38.39% and 38.60%, respectively.

The phylogenetic tree of *NPC1* amino acid sequences was constructed as shown in Figure 1. Among the crustaceans, the *NPC1s* from the *Penaeus* spp. were the first to cluster, followed by those of the crayfish, Brachyura, and *M. nipponense*. The insect *NPC1s* clustered into one group and then into a larger branch with the crustaceans. The vertebrate *NPC1s* were grouped into one branch.

### 2.3. Expression Analysis of Mn-NPC1

The results of *Mn-NPC1* expression in the different tissues of females are shown in Figure 2A. *Mn-NPC1* was expressed in every tissue examined: weakly in the eyestalks, gills, cranial ganglia, and muscle; moderately in the heart and ovary; and extremely highly in the hepatopancreas (*p* < 0.05).

We further investigated the expression of *Mn-NPC1* in the embryonic and larval stages (Figure 2B). The qPCR CT results indicated that *Mn-NPC1* was hardly expressed in any of the embryonic developmental stages (from the CS–ZS stages) (*p* > 0.05). It began to express weakly on the 1st day after hatching (L1) and maintained a very low level of expression without significant variation until the 10th day after metamorphosis (PL10) (*p* > 0.05). The level of *Mn-NPC1* expression was significantly up-regulated on the 15th day after metamorphosis (PL15) (*p* < 0.05) but decreased at PL20 (*p* < 0.05). Its expression level was sharply up-regulated on the 25th day after metamorphosis (PL25) (*p* < 0.05).

The expression profiles in the hepatopancreas and ovary were further analyzed during the five ovarian developmental stages. *Mn-NPC1* expression in the ovary was positively correlated with ovarian development (Figure 2C). Its expression increased from stages I–IV and then decreased significantly from stages IV–V. Based on the qPCR CT analysis, *Mn-NPC1* was only very weakly expressed in stages I, II, and III, with only slow and non-significant changes (*p* > 0.05). There was a sharp, nearly 20-fold, increase in expression in the mature stage (stage IV) (*p* < 0.01), followed by a sharp decrease in stage V (*p* < 0.01). Its expression pattern in the hepatopancreas (Figure 2D) was significantly different, being relatively higher in stages I and III (*p* < 0.05), but relatively lower, and not significantly different, in the other stages (*p* < 0.05).

### 2.4. Subcellular Localization of Mn-NPC1

In situ hybridization (ISH) was used to detect the localization of *Mn-NPC1* in the ovarian stages (Figure 3). The ISH results showed that the *Mn-NPC1* signals were generally weak in all of the ovarian developmental stages. *Mn-NPC1* distribution was observed in the cytoplasmic membranes, nuclei, and follicle membranes, with weak signals. It was mainly located in the cytoplasmic membranes and nuclei of the oocytes. The results also showed that the *Mn-NPC1* signal was strongest in the mature ovary stage, while it was weakest in the immature stages.

### 2.5. Functional Analysis of Mn-NPC1 in Ovarian Maturation Regulation

RNAi technology was used to investigate the function of *Mn-NPC1* in regulating ovarian maturation. The short-term interference results showed that the expression of *Mn-NPC1* in the experimental group, in which *Mn-NPC1* was knocked down using dsRNA, was almost the same as in the control group after the first day (*p* > 0.05). Compared with the control, *Mn-NPC1* expression in the experimental group decreased by 70.24% on the fourth day (*p* < 0.05) and by 92.18% on the seventh day (*p* < 0.01) (Figure 4).

The long-term experiment lasted for 25 days with dsRNA injections given every five days to keep *Mn-NPC1* expression at a very low level. Daily observations were made to record the ovarian developmental stage and molting frequency, and the gonadosomatic index (GSI) and ecdysterone content were also calculated.

The results showed that the GSIs did not differ significantly between the experimental and control groups on the first and ninth days (*p* > 0.05). On the 17th day, the GSI of the experimental group differed significantly from the control group (*p* < 0.05). On the 25th day, the difference in GSI between the experimental and control groups increased significantly (*p* < 0.05) (Figure 5A). We also recorded the cumulative proportion of stage III ovaries in the two groups (Figure 5B). On the first day, none of the prawns in either group had stage III ovaries (*p* > 0.05). On the ninth day, 7.02% and 5.97% of the prawns in the control and experimental groups had stage III ovaries, respectively (*p* > 0.05). From the 17th day, the cumulative proportion of stage III ovaries began to show significant differences between the two groups (*p* < 0.05). On the last sample day (25th day), 20.75% and 13.64% of the prawns had stage III ovaries in the control and experimental groups, respectively (*p* < 0.05). At the end of the long-term RNAi experiment, most prawn ovaries were in stage II or were about to enter stage III. Stage II ovaries from both groups were chosen at random for paraffin section observations (Figure 5C). There was no difference in ovarian structure between the two groups, indicating that dsRNA suppressed ovarian maturation rate, rather than causing structural deformities.

Cumulative molting frequency statistics showed no differences between the two groups on the first and ninth days (*p* > 0.05), but a significant difference in cumulative molting frequency had appeared by the 17th day (*p* < 0.05). By the 25th day, the difference between the two groups had increased (*p* < 0.05) (Figure 6A). The ecdysterone contents were determined using an ELISA test (Figure 6B). The results showed that the ecdysterone content gradually increased with the development of the ovary and that there were no differences in ecdysterone content between the two groups on the 1st, 9th, and 17th days (*p* > 0.05). By the 25th day, the ecdysterone content in the experimental group was significantly lower than in the control group (*p* < 0.05).

## 3. Discussion

Sterols are structural compounds that regulate membrane permeability and fluidity, as well as have metabolic effects in animals including insects [25]. In mammals, it has been shown that *NPC1* and *NPC2* in the Niemann–Pick C-type protein family are important players in sterol metabolism and transport processes, regulating sterol and other lipid levels [17,26]. Sterols are also important precursors of insect steroid hormones, such as ecdysone, which regulate molting [27,28]. Cholesterols are important sterols in insects and are the most basic precursors of insect ecdysone, playing a key role in regulating the reproductive and molting processes [29,30]. However, there is little research into *NPC1* and *NPC2* in insects and the small number of studies undertaken have mainly focused on model species such as *D. melanogaster* and *B. mori*, with no studies into crustaceans except for some sequences listed on the NCBI database. In our previous study, we performed a differentially expressed protein analysis and KEGG enrichment analysis of the hepatopancreas proteome of female *M. nipponense* during five of the ovarian developmental stages and screened for the *NPC1* protein and its significantly enriched lysosome pathways [13]. Therefore, our functional analysis of *NPC1* in regulating ovarian maturation in crustaceans is the first of its kind.

Two homologous NPC1s (*NPC1a* and *NPC1b*) have been reported in *H. sapiens*, *Mus musculus* and *D. melanogaster*, and were found to be responsible for dietary cholesterol absorption and intracellular cholesterol transport, respectively [31]. However, other insects, such as *B. tabacis*, have been reported to only have NPC1, which may be involved in both of the above-mentioned physiological processes [24]. In this study, only one NPC1 homolog was found in *M. nipponense*, which exhibited three typical conserved domains of the NPC family (NPC1_N, SSD, and PTC). *Mn-NPC1* contained 12 typical transmembrane domains, less than the 13–15 found in various insect species. An alignment analysis showed that the amino acid similarity between *Mn-NPC1* and other crustacean *NPC1s* was greater than 50% and that the amino acid similarity between *Mn-NPC1* and vertebrate *NPC1s* was nearly 40%. All of these results indicate that NPC1 is functionally conserved over evolutionary time.

*NPC1* was reported to be widely distributed in all of the tissues examined in other species [20,23]. It was highly expressed in mammalian intestinal epithelial cells [32]. In *Drosophila* spp., *NPC1a* had a wide tissue distribution, while *NPC1b* was only expressed in the midgut epithelial cells [31]. The highest expression of *NPC1* in *B. mori* was found in the testes, while, in *B. tabaci*, it was in the abdomen [24]. These results suggest that *NPC1* may play different crucial physiological roles in different species. In the crustacean *M. nipponense*, the distribution of *Mn-NPC1* was significantly different from the other species studied. *Mn-NPC1* tissue expression indicated that crustacean *NPC1* may be mainly involved in the transport of sterols in the hepatopancreas, ovaries, and heart. In the insects *B. mori* and *B. tabaci*, expression of *NPC1* was found in all of the larval stages and was highest in the early larval stages and in adults, suggesting that it not only plays a key role in development but is also an essential participant in adult-specific physiological activities [23,24]. In the crustacean *M. nipponense*, *Mn-NPC1* was almost absent during embryonic development and the early larval developmental stages and was up-regulated until the 15th day after metamorphosis, suggesting that it was not involved in the developmental stages, but only played a key role in adults. Studies on *B. mori* showed that *NPC1* was highly expressed in both the testes and ovaries, while *NPC1a* was shown to be essential for spermatogenesis in *Drosophila* spp. [22,23]. In this study, the expression patterns of *Mn-NPC1* in the hepatopancreas and ovary were further examined, and its expression was found to be relatively stable in the hepatopancreas but was positively correlated with maturation in the ovaries.

The function of *NPC1* in some insects has also been demonstrated by RNAi. In *B. mori*, the knockdown of *NPC1* led to cholesterol accumulation in cells and also affected larval molting [23]. In *B. tabaci*, RNAi knockdown of *NPC1* resulted in a decrease in female fecundity and an increase in larval mortality [24]. Insect ecdysone, which regulates insect molting behavior, is crucial for growth in insects. Based on studies of the Halloween gene family, insect ecdysone can regulate reproduction [33]. The insect ecdysone that regulated molting behavior was found to be a sterol. The RNAi results above confirm that *NPC1* plays an important role in sterol metabolism and transport, indirectly promoting insect molting and ovarian maturation by the regulation of ecdysone synthesis. We used RNAi to study the function of *Mn-NPC1* in crustaceans. Our long-term interference experiment results showed that *Mn-NPC1* knockdown significantly reduced the GSI and ecdysone contents of female *M. nipponense*. In contrast with our previous study on the Halloween family genes (such as CYP307 A1, CYP302 A1, and CYP306A1), knockdown of *Mn-NPC1* had a lesser effect on ovarian maturation delay and decrease of molting frequency than knockdown of the Halloween family genes [34,35,36]. These results suggest that while *NPC1* is an important regulator of sterol transport in *M. nipponense*, it is not the only factor involved. Knockdown of *Mn-NPC1* led to a decrease in ecdysone levels, leading to delayed ovarian maturation and reduced molting frequency.

In the breeding seasons of *M. nipponense*, molting occurs simultaneously with ovarian maturation, which is known as reproductive molting [34]. Our results suggested that Mn-NPC1 is crucial in controlling the reproductive molting process in female *M. nipponense* and its depletion slows down ovarian maturation and molting activities. Similar to insects, crustaceans cannot synthesize cholesterol itself and require exogenous cholesterol by feed [35]. Therefore, the exogenous cholesterol supplementation level may have a significant effect on the expression of NPC. In prawn aquaculture, a reasonable dose of cholesterol in feed can regulate NPC expression, as well as promote ovarian maturation and molting. The regulatory mechanisms of NPC1 in ovarian maturation, such as the synthesis site in the hepatopancreas and the cholesterol transport routes, need to be elucidated in further research.

## 4. Materials and Methods

### 4.1. Experimental Prawns and Sample Collection

All of the experimental *M. nipponense* prawns were obtained from the Dapu Scientific Experimental Base, the Freshwater Fisheries Research Center (Wuxi, China) between May and August 2023. Thirty healthy adult female *M. nipponense* prawns with body weights of 1.55 ± 0.22 g were used for the tissue expression study. The various tissues were dissected out and placed into liquid nitrogen. The tissues tested included the ovary, cerebral ganglion, muscles, heart, eyestalks, gills, and hepatopancreas. Each tissue sample was composed of five biological replicates and one replicate contained three randomly selected prawns. Fifty adult female *M. nipponense* prawns (body weight ± SD: 1.78 ± 0.36 g) with ovaries at different developmental stages were used for the ovarian expression profile study. The five ovarian developmental stages were defined by ovary color according to a previous study [37]. Each sample of each stage contained five biological replicates and one replicate contained three randomly selected prawns. The stage categories of embryos were also defined according to a previous study [37]. Assignment of the five embryonic developmental stages (cleavage, CS; blastula, BS; gastrulation, GS; nauplius, NS, and zoea, ZS) was based on microscopic observations. Each stage sample contained five biological replicates and one replicate contained 20 randomly selected embryos. Sampling of the larval and post-larval developmental stages (L and PL stages) was performed every five days. The samples were taken at nine points: L1, L5, L10, L15, PL1, PL5, PL10, PL15, and PL25. Each stage sample contained five biological replicates and one replicate contained 10 randomly selected larvae. All of the samples were stored in liquid nitrogen.

### 4.2. RNA Extraction and Amplification of Mn-NPC1

Extractions of the total RNA and cDNA were performed using RNAiso Plus Reagent and an RNA PCR Kit (Code No. 9109, TaKaRa, Tokyo, Japan) according to the manufacturer’s instructions. The partial *Mn-NPC1*cDNA sequence of *M. nipponense* was obtained from the proteome database (http://proteomecentral.proteomexchange.org, PXD037141, accessed on 10th April 2023) [13]. Three pairs of primers were designed for validation. The PCR conditions followed those of a previous study [23]. The processes of 3′ and 5′ cloning were performed using a RACE PCR Kit (Code No.634858, TaKaRa, Shiga, Japan). The primers used are listed in Table 1. The PCR products were sequenced by Sangon Biotech (Shanghai, China).

### 4.3. Bioinformatics Sequence Analysis

The website https://services.healthtech.dtu.dk/service.php?SignalP-3.0 (accessed on 11 April 2023) was used to predict the signal peptides. The website https://web.expasy.org/compute/ (accessed on 11 April 2023) was used to analyze the protein molecular weights and predict the isoelectric points. The website https://dtu.biolib.com/DeepTMHMM (accessed on 11 April 2023) was used to predict the transmembrane region and N-glycosylation site. The website http://smart.emblheidelberg.de/ (accessed on 11 April 2023) was used to predict the various domains. DNAMAN 6.0 (https://dnaman.software.informer.com/6.0/) (accessed on 11 April 2023) was used for sequence alignment and MEGA5.0 (https://www.megasoftware.net/) (accessed on 11 April 2023) for phylogenetic tree construction based on amino acid sequences. The amino acid sequences of the other animals used in this study were obtained from http://www.ncbi.nlm.nih.gov/ (accessed on 11 April 2023).

### 4.4. In Situ Hybridization

ISH was used to detect the subcellular localization of the *Mn-NPC1s* in the five ovarian developmental stages. Ovaries of five different developmental stages were dissected out and soaked in 4% paraformaldehyde (PBS, pH 7.4) for 12 h at 4 °C. A Zytofast PLUS CISH implementation kit (Code No.T-1061-40, Zyto Vision GmBH, Bremerhaven, Germany) was used for tissue fixation and paraffin embedding. Both sense and antisense probes with digoxin signals were synthesized by Sangon Biotech (Table 1). Hematoxylin-eosin staining was used for the blank control group, while, for the negative control group, we used the antisense probe. All slides were examined using a light microscope.

### 4.5. RNA Interference

RNAi technology was used to study the function of *Mn-NPC1* in the regulation of ovarian maturation. The *GFP* gene was used as a control gene [38]. RNAi primers for *Mn-NPC1* and *GFP* were designed using the website http://www.flyrnai.org/cgi-bin/RNAi_find_primers.pl (accessed on 15 July 2023) and are listed in Table 1. A Transcript AidTMT7 high-yield transcription kit (Code No. K0441, Fermentas, Inc., Rockville, MD, USA) was used to synthesize dsRNA.

Most of the female ovaries were in stage I when the study began and were used in the RNAi experiment. For the short-term RNAi experiment, ninety healthy female prawns (1.42 ± 0.19 g) in ovary stage I were selected and randomly divided into two groups: a control group and an experimental group. Each group contained three parallel lines (n = 15) and was kept in tanks at a water temperature of 24 ± 1 °C. Prawns in the control and experimental groups were injected with ds-GFP and ds-NPC1, at a dose of 4 μg/g body weight (b.w.), respectively. Nine prawns (three biological replicates) were randomly sampled from each group on the first, fourth, and seventh day after injection, and their ovaries were dissected out for the interference efficiency investigation.

In the long-term RNAi experiment, 180 healthy female prawns (1.57 ± 0.22 g) in ovary stage I were selected and randomly divided into two groups: a control group and an experimental group. Each group contained three parallel lines (n = 30) and was kept in tanks at a water temperature of 24 ± 1 °C). Every five days, the prawns in the control and experimental groups were injected with ds-GFP and ds-NPC1 at a dose of 4 μg/g b.w., respectively. Nine prawns (three biological replicates in each parallel line) were randomly sampled from each group on the 1st, 9th 17th, and 25th day after injection, and their ovaries were dissected out and measured. Both body weight and ovary weight were recorded to calculate the GSI (GSI = gonad weight/body weight × 100%). Paraffin sections were used to compare the experimental and control ovaries at the end of the study. The dissected ovaries were then used for ecdysterone content determination using a Penaeus Ecdysterone ELISA Kit (Code No. ml036262, Mlbio, Shanghai, China). The developmental stage of the ovary of each prawn was recorded daily according to the criteria of a previous study [34]. The molting frequency of each parallel line was also recorded daily to calculate the cumulative number of molts.

### 4.6. qPCR and Statistical Analysis

The *EIF* gene (eukaryotic translation initiation factor 5A) was used as the reference gene (Table 1). The q-PCR conditions followed those of a previous study [34]. The expression levels were calculated following the 2^−ΔΔCT^ method [39]. All quantitative data conformed to the homogeneity of variance and the normal distribution, and all the data were presented as mean ± standard deviation. One-way ANOVA and two-tailed *t*-tests in SPSS 23.0 were used in the statistical analyses. Significant differences were indicated by *p* < 0.05.

## 5. Conclusions

This is the first study of *NPC1* in crustaceans. The sequence results suggest that *Mn-NPC1* is functionally conserved with other vertebrates and invertebrates. In contrast to insects, the *Mn-NPC1* expression results show that *Mn-NPC1* plays a critical biological role in adults, but not in embryos or larvae. In *M. nipponense*, *Mn-NPC1* was mainly distributed in the hepatopancreas of adult females, and its expression was positively correlated with ovarian development. Furthermore, it played important biological roles in both ovarian maturation and molting in *M. nipponense* by regulating ecdysone. The results of this study expand our knowledge of *NPC1* in crustaceans and provide further information regarding the regulatory mechanisms of ovarian maturation in *M. nipponense*. The regulatory mechanisms of NPC1 in ovarian maturation, such as the synthesis site in the hepatopancreas and cholesterol transport routes, need to be elucidated in further research.

## Figures and Tables

**Figure 1 ijms-25-06049-f001:**
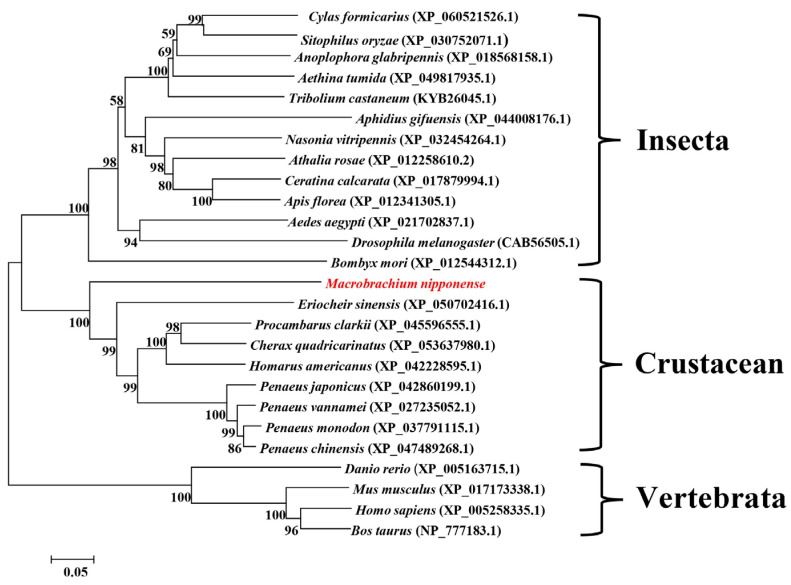
Phylogenetic tree of *Mn-NPC1* amino acid sequence in different groups. Red font indicated the *Mn-NPC1* amino acid sequence.

**Figure 2 ijms-25-06049-f002:**
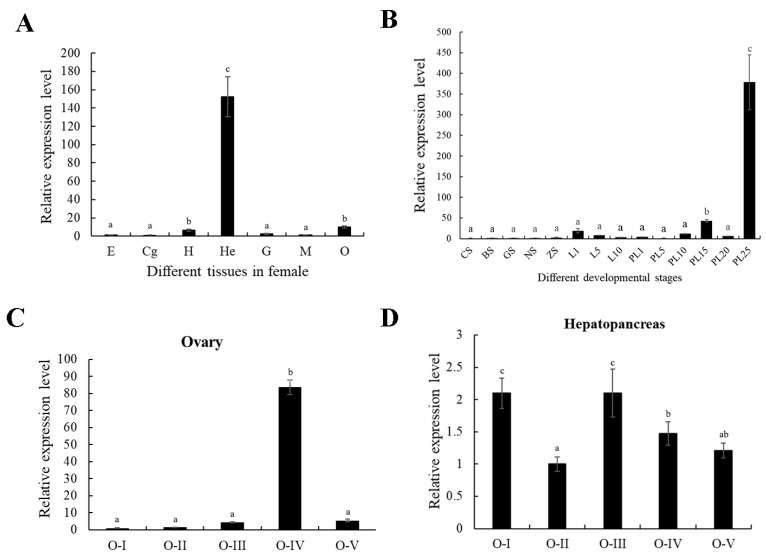
Expression analysis of *Mn-NPC1*. (**A**) Tissue distribution in females; (**B**) expression pattern in developmental stages; (**C**) expression patterns in ovary at different ovarian stages; (**D**) expression pattern in the hepatopancreas at different ovarian stages. (**A**) E, eyestalk; Cg, cerebral ganglion; H, heart; He, hepatopancreas; G, gill; M, muscle; O, ovary. (**B**) CS, cleavage stage; BS, blastocyst stage; GS, gastrulation stage; NS, nauplius stage; ZS, zoea stage; L1, the first-day larvae after hatching; L5, the fifth-day larvae after hatching; L10, the 10th-day larvae after hatching; L15, the 15th-day larvae after hatching; PL1, post-larval stage of the first day; PL5, post-larval stage of five days; PL10, post-larval stage of ten days; PL15, post-larval stage of 15 days; PL-25, post-larval stage of 25 days. (**C**,**D**): different ovarian stage expressions, OI undeveloped stage, OII developing stage, OIII nearly-ripe stage, OIV ripe stage, OV spent stage. Different lowercase letters indicate significant differences (*p* < 0.05).

**Figure 3 ijms-25-06049-f003:**
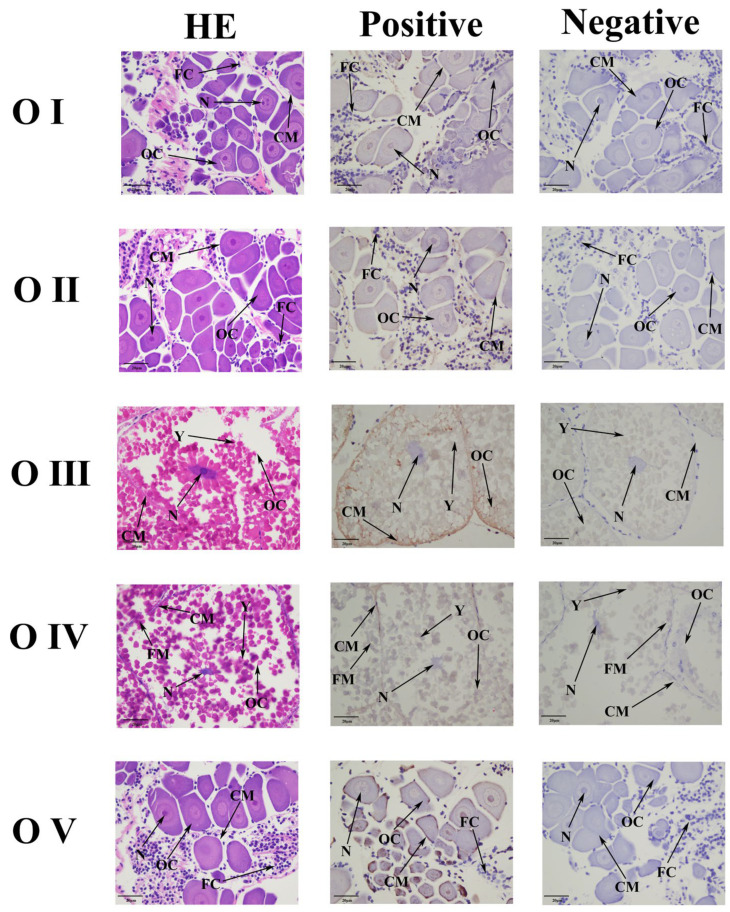
Location of *Mn-NPC1* detected in the ovary by in situ hybridization. (OC) Oocyte; (N) nucleus; (CM) cytoplasmic membrane; (Y) yolk granule; (FC) follicle cell; (FM) follicle membrane. Scale bars: ×400.

**Figure 4 ijms-25-06049-f004:**
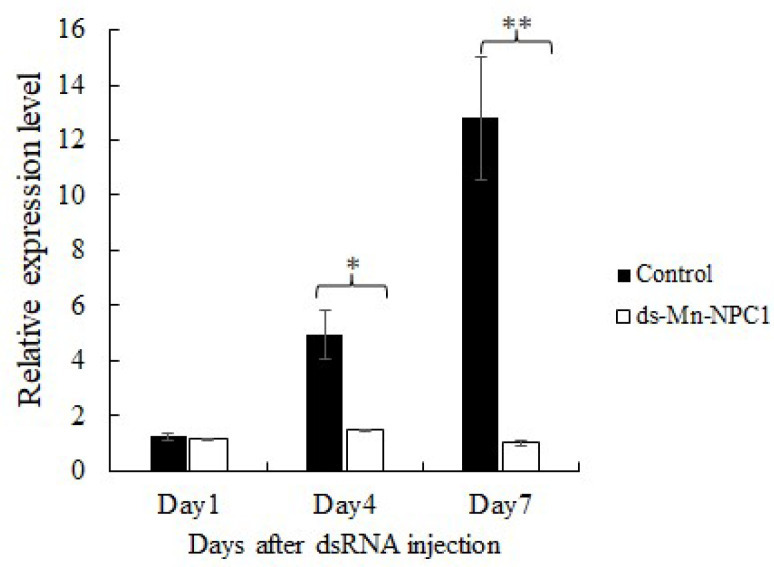
Efficiency of RNAi of *Mn-NPC1* in ovary. * denotes statistical significance of *p* < 0.05 and ** denotes statistical significance of *p* < 0.01.

**Figure 5 ijms-25-06049-f005:**
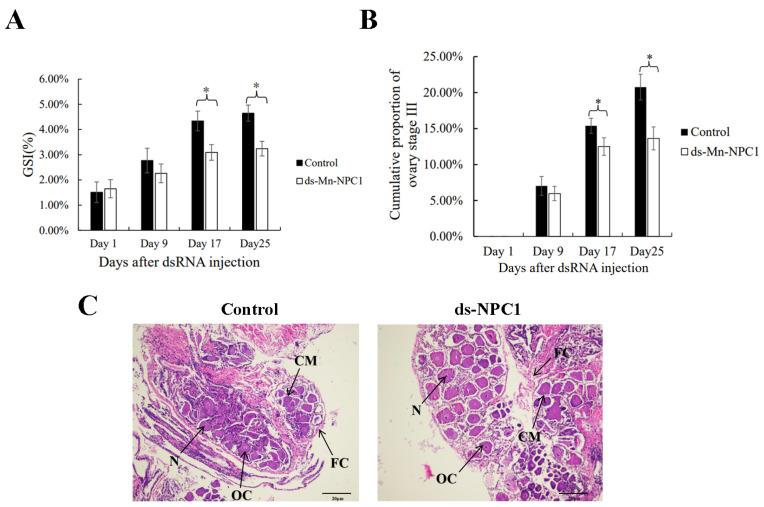
Function of *Mn-NPC* in ovary. (**A**) Changes of GSI (%) of female *M. nipponense* after RNAi; (**B**) changes of cumulative proportion of ovary stage III of female *M. nipponense* after RNAi; (**C**): ovary paraffin section in two groups after RNAi; * denotes statistical significance of *p* < 0.05. (OC) Oocyte; (N) nucleus; (CM) cytoplasmic membrane; (Y) yolk granule; (FC) follicle cell; (FM) follicle membrane. Scale bars: ×100.

**Figure 6 ijms-25-06049-f006:**
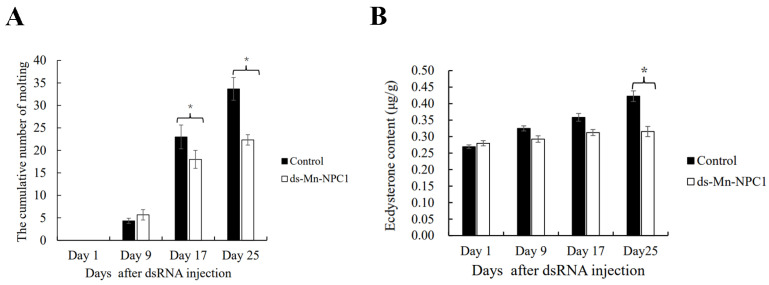
Function of *Mn-NPC* in molting. (**A**) Changes of cumulative molting number in female *M. nipponense* after RNAi; (**B**) changes of 20E content in female *M. nipponense* after RNAi. * denotes statistical significance of *p* < 0.05.

**Table 1 ijms-25-06049-t001:** Primers used in this study.

Primer	Primer Sequence (5′-3′)
*Mn-NPC1* F1(ORF)	CTACTCTGTTTCTTTTGGCGAC
*Mn-NPC1* R1(ORF)	TAGGTGACAGAGGAGCATAGCA
*Mn-NPC1* F2(ORF)	GATATCGTCCACCTCGGAGACA
*Mn-NPC1* R2(ORF)	ATGACGCTGGCTGGGCGATGTA
*Mn-NPC1* F3 (ORF)	CTTCTATGATACCGAACGTGAC
*Mn-NPC1* R3(ORF)	TTACCAGAAACATTGTCATCAG
*Mn-NPC1* R (5′)	AGCAGGCTTGGGTGGGCCGTTG
*Mn-NPC1* F (3′)	CCTGAAAGTCTTCAAGAATCAA
*Mn-NPC1* F(qPCR)	GATCCTACACTAAGCATCCCCTG
*Mn-NPC1* R(qPCR)	GATCCTACACTAAGCATCCCCTG
*EIF*-F (qPCR)	CATGGATGTACCTGTGGTGAAAC
*EIF*-R (qPCR)	CTGTCAGCAGAAGGTCCTCATTA
*Mn-NPC1* probe	CGGTTTCTCGCCTTTTTCATTTGATTTTGTCTATG
*Mn-NPC1* anti-probe	CATAGACAAAATCAAATGAAAAAGGCGAGAAACCG
*Mn-NPC1* iF (RNAi)	TAATACGACTCACTATAGGGGTCGTGGCTCTCTCTTACGG
*Mn-NPC1* iR (RNAi)	TAATACGACTCACTATAGGGGACACTTTGTGTTGCGCAGT
*GFP* iF (RNAi)	GATCACTAATACGACTCACTATAGGGTCCTGGTCGAGCTGGACGG
*GFP* iR (RNAi)	GATCACTAATACGACTCACTATAGGGCGCTTCTCGTTGGGGTCTTTG

## Data Availability

Data information will be provided upon request.

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
