# Peer review of "NPC Intracellular Cholesterol Transporter 1 Regulates Ovarian Maturation and Molting in Female Macrobrachium nipponense"

_ijms, 2024, doi:10.3390/ijms25116049_

Round 1

Reviewer 1 Report (Previous Reviewer 2)

Comments and Suggestions for Authors

The auhthors have provided the cDNA sequence  of NPC1 in M. nipponense (Mn-NPC1). The expression profiles and subcellular localization in various tissues and at various developmental stages were perforemd. Its function in regulating ovarian maturation in M. nipponense  was partly examined. This is straight forward study which deserves to be considered for publication after some revision

The number of animals should be given in all experiments and also indicate whether true biological replicates were made!

In Fig. 3 with the ISH results there is a need to enlarge this photos so that its is possble to see whether the NPC is anchored on the membranes or not since it has so many TM domains?

In Figure 5 B the  difference between the control and the silenced animals are not impressive and one must ask whether this is a true difference and to consolidate this how many individual animals were used in each experiment? This has to be detailed?

One question is why the NPC was so avidly expressed in hepatopancreas (Fig. 2D) and not very much in most developmental stages in ovary?

Why was not ISH performed on hepatopancreas cells which contained most of the expressed  NPC? Maybe this could give a clue as to why it is so highly expressed there? Are all cells containing NPC or can some distinct cells be shown to have this NPC for example?

Comments on the Quality of English Language

OK with some minor editing

Author Response

Dear Reviewer 1,

First of all, we are very grateful for your recognition of our research work. Thank you very much for your comments and suggestions. The valuable comments from you not only helped us with the improvement of our manuscript, but suggested some ideas for future studies.

Below you will find our responses to your comments:

  1. The number of animals should be given in all experiments and also indicate whether true biological replicates were made!

Response: Thanks for your suggestion. We add details about the number of animals in all experiments. For tissue expression study, each tissue sample was composed of five biological replicates and one replicate contained three randomly selected prawns. For the ovarian expression profile study, each sample of each stage contained five biological replicates and one replicate contained three randomly selected prawns. For the embryos expression profile study, each stage sample contained five biological replicates and one replicate contained 20 randomly selected embryos. For the larval and post-larval developmental stages expression profile study, each stage sample contained five biological replicates and one replicate contained 10 randomly selected larval.

  1. In Fig. 3 with the ISH results there is a need to enlarge this photos so that its is possble to see whether the NPC is anchored on the membranes or not since it has so many TM domains?

Response: The in situ hybridization technology use isotope-labeled nucleic acid probes to localize genes in cells or tissues, and it can not show the anchoring of proteins domains and cell membranes. We attached the vector diagram of Figure 3 with 600 dpi to show the position clearly.

  1. In Figure 5 B the  difference between the control and the silenced animals are not impressive and one must ask whether this is a true difference and to consolidate this how many individual animals were used in each experiment? This has to be detailed?

Response: First of all, thanks for your kindly reminding. We re-examined Figures 5A and 5B and found that the titles of y-axis was wrong which should be percentage values for GSI and the cumulative proportion of ovary stage â…¢. We corrected this in new Figure 5 in the revised manuscript.

In the long-term RNAi experiment, 180 healthy female prawns (1.57 ± 0.22 g) in the ovary stage I were selected and randomly divided into two groups: a control group and an experimental group. Each group contained three parallel lines (n = 30) and were kept in tanks at a water temperature of 24 ± 1°C). Every five days, the prawns in the control and experimental groups were injected with ds-GFP and ds-NPC1, at a dose of 4 μg/g b.w., respectively. Nine prawns (three biological replicates in each parallel line) were randomly sampled from each group on the 1st, 9th 17th, and 25th day after injection, and their ovaries were dissected out and measured. Both body weight and ovary weight were recorded to calculate the GSI (GSI = gonad weight/body weight × 100%). Paraffin sections were used to compare the experimental and control ovaries at the end of the study. The dissected ova-ries were then used for ecdysterone content determination. There were statistically significant differences in these parameters by two-tailed t-test.

  1. One question is why the NPC was so avidly expressed in hepatopancreas (Fig. 2D) and not very much in most developmental stages in ovary?

Response: the reason that why NPC was so avidly expressed in hepatopancreas and weakly expressed in ovary might be the hepatopancreas is the main organ for cholesterol metabolism and NPC is the free cholesterol transporter.

  1. Why was not ISH performed on hepatopancreas cells which contained most of the expressed  NPC? Maybe this could give a clue as to why it is so highly expressed there? Are all cells containing NPC or can some distinct cells be shown to have this NPC for example?

Response: Thanks for your good suggestion. Hepatopancreas, in crustaceans, is an important site for vitellogenin synthesis, which provides energy for ovary maturation, as well as essential fatty acids and cholesterol for sex steroid hormones synthesis. In previous study, comparative transcriptomic analysis of hepatopancreas during five different ovarian stages were proceeded and many the differential genes (DEGs) which related to ovarian maturation were screened. NPC1 was one of the candidate ovarian maturation related genes. The aim of this study was mainly focused on the function of NPC1 in regulating ovarian maturation and the results proved it played promoting roles in ovarian maturation. In further researches, we will study the regulating mechanism of NPC1in ovarian maturation including its synthesis site in the hepatopancreas, transport routes and so on.

Reviewer 2 Report (Previous Reviewer 1)

Comments and Suggestions for Authors

General comments

In this paper, the authors investigated the cloning of a cDNA encoding NPC intracellular cholesterol transporters 1 (Mn-NPC1) from the oriental river prawn Macrobrachium nipponense. Mn-NPC1 was highly expressed in the hepatopancreas, moderately in the ovary and heart. Expression levels of Mn-NPC1 in the ovary positively correlated with vitellogenesis. Gene silencing of Mn-NPC1 reduced GSI and 20-hydroxyecdysone (20E) content in the ovary. Therefore, the authors suggest that Mn-NPC1 may control ovarian maturation. There are, however, some serious issues described below, and therefore this article at this form might not be acceptable for the publication in International Journal of Molecular Sciences.

Specific comments

1) The authors showed the gene expression of Mn-NPC1 only in females. It is not clear why it was not examined in males.

2) Despite no relation between 20E content in the ovary and molting, why the authors measured 20E in the ovary rather than in the hemolymph? Additionally, information on the ELISA kits used in this study was not obtained through my search. The web page or manufacture’s instruction should be provided.

Author Response

Dear Reviewer 2,

First of all, we are very grateful for your recognition of our research work. Thank you very much for your comments and suggestions. The valuable comments from you not only helped us with the improvement of our manuscript, but suggested some ideas for future studies.

Below you will find our responses to your comments:

  1.  The authors showed the gene expression of Mn-NPC1 only in females. It is not clear why it was not examined in males.

 Response: As we wrote in the “Introduction”, the purpose of this study was mainly focused on the exploration the potential mechanisms of fast ovary development in M. nipponense. In previous research of our team, comparative hepatopancreatic proteomic analysis of female prawn during five ovarian maturation stages were performed to explore differentially expressed proteins and pathways regulating ovarian maturation. An NPC1 protein was found to be listed as the top up-regulated protein in vitellogenesis stage (ovary stage II to III) and top down-regulated protein in emptying stage (ovary stage IV to V), indicating its significant roles in regulating ovarian maturation. Therefore, this study was aimed to investigate the function of Mn-NPC1 in ovarian maturation. The function of Mn-NPC1 in male will be studied in further researches.

  1. Despite no relation between 20E content in the ovary and molting, why the authors measured 20E in the ovary rather than in the hemolymph? Additionally, information on the ELISA kits used in this study was not obtained through my search. The web page or manufacture’s instuction should be provided.

Response: Thanks for your kind suggestions. In crustaceans, free ecdysteroids including 20-hydroxyecdysone and ecdysone have been proved to be present in both ovaries and hemolymph (Young et al., 1993a; Young et al., 1993b). There are also studies to prove that haemolymph ecdysteroid titres relate to changes in molting status but not to changes in oocyte development and the ovary are capable of de novo synthesis of the ecdysteroids (Styrishave et al., 2008). In this study, we mainly focused on the potential roles of Mn-NPC1 in ovarian maturation in M. nipponense.  Therefore, we measured the ecdysteroids content in the ovary. According to your kindly reminder, we find that we made mistakes in our presentation of ecdysteroids measurement. We used the Penaeus Ecdysterone ELISA Kit to measure the ecdysterone content. We revised the presentation of ecdysteroids in the manuscript for ecdysterone content. The manufacture’s instruction of the Kit is attached in revised package.

Young, N. J., Webster, S. G., & Rees, H. H. (1993) a. Ovarian and hemolymph ecdysteroid titers during vitellogenesis in Macrobrachium rosenbergii. General and comparative endocrinology, 90(2), 183-191.

Young, N. J., Webster, S. G., & Rees, H. H. (1993) b. Ecdysteroid profiles and vitellogenesis in Penaeus monodon (Crustacea: Decapoda). Invertebrate reproduction & development, 24(2), 107-117.

Styrishave, B., Lund, T., & Andersen, O. (2008). Ecdysteroids in female shore crabs Carcinus maenas during the moulting cycle and oocyte development. Journal of the Marine Biological Association of the United Kingdom, 88(3), 575-581.

Reviewer 3 Report (New Reviewer)

Comments and Suggestions for Authors

Abstract

This section is well prepared. I recommend that the authors briefly include your experimental method for ovarian maturation in female prawns.

Introduction

This section is well prepared.

Material and methods

Line 293 “were defined by ovary color according to a previous study [REFERENCE]. Each sample” I think you missed one reference. Please check. Same problem in line 295.

Include the catalog number of each kit that you used.

Did you evaluate the Normality and Homogeneity of the data? This information is essential for parametric analysis. Include this information.

Results

This section must be revised carefully.

The letter assignation in Figure 1A and D is rare because you are indicating that He is “a” and H and O are “c” and E, Cg, G, and M are “b” but they have the lowest values (Fig. 1A).

The statistical analyses must be reviewed in the MS.

Discussion

This section is well prepared.

I have some reflective questions for improving this section.

Under what environmental conditions could NPC1 expression be affected?

Is it possible to reverse the effects that affect NPC1 expression and synthesis?

Briefly include a subsection that resolves these questions.

Conclusion

Adjust this section according to my previous comments.

Author Response

Dear Reviewer 3,

First of all, we are very grateful for your recognition of our research work. Thank you very much for your comments and suggestions. The valuable comments from you not only helped us with the improvement of our manuscript, but suggested some ideas for future studies.

Below you will find our responses to your comments:

  1. Abstract, this section is well prepared. I recommend that the authors briefly include your experimental method for ovarian maturation in female prawns.

 Response: Thanks for your good comments. We add a brief sentence in Abstract. Please find in the revised manuscript. “A 25day RNA interference experiment was employed to illustrate the Mn-NPC1 function in ovary maturation. Experimental knockdown of Mn-NPC1 using dsRNA resulted in a marked reduction in the gonadosomatic index and ecdysone content of M. nipponense females”.

  1. Introduction, this section is well prepared.

 Response: Thanks for your comments.

  1. Material and methods, Line 293 “were defined by ovary color according to a previous study [REFERENCE]. Each sample” I think you missed one reference. Please check. Same problem in line 295.

 Response: Thanks for your kindly reminding. We add the missed reference in the revised manuscript.

Qiao, H.; Xiong, Y.W.; Zhang, W.Y.; Fu, H.T.; Jiang, S.F.; Bai, H.K.; Sun, S.M.; Jin, S.B.; Gong, Y.S. Characterization, expression, and function analysis of gonad-inhibiting hormone in Oriental River prawn, Macrobrachium nipponense and its induced expression by temperature. Comp. Biochem. Phys. A 2015, 185, 1-8.

  1. Include the catalog number of each kit that you used.

 Response: Thanks for your kindly reminding. We add the catalog number of each kit in the revised manuscript.

  1. Did you evaluate the Normality and Homogeneity of the data? This information is essential for parametric analysis. Include this information.

 Response: Thanks for your suggestion. We evaluated the Normality and Homogeneity of the data before parametric analysis. We add this information in the revised manuscript.

  1. The letter assignation in Figure 1A and D is rare because you are indicating that He is “a” and H and O are “c” and E, Cg, G, and M are “b” but they have the lowest values (Fig. 1A).

  Response: Thanks for your kindly reminding. I think you mean the Figure 2 (Expression analysis of Mn-NPC1).

We revised the letter assignation in Figure 1.

  1. The statistical analyses must be reviewed in the MS.

Response: Thanks for your kindly reminding. We reviewed statistical analyses in the revised manuscript.

  1. Discussion, this section is well prepared. I have some reflective questions for improving this section.

 Response: Thanks for your comments.

  1. Under what environmental conditions could NPC1 expression be affected? Is it possible to reverse the effects that affect NPC1 expression and synthesis? Briefly include a subsection that resolves these questions.

  Response: Thanks for your good comments. We have added a brief subsection to resolve these questions at the end of the discussion.

In breeding seasons of M. nipponense, the molting is occurred simultaneously with the ovarian maturation, which was known as reproductive molting [34]. Our results suggested that Mn-NPC1 was crucial to control the reproductive molting process in female M. nipponense and its depletion slowed down the ovarian maturation and molting activities. Similar to insects, crustaceans cannot synthesize cholesterol itself and require exogenous cholesterol by feed [35]. Therefore, exogenous cholesterol supplementation level may had a significant effect on the expression of NPC. In the prawn aquaculture, reasonable dose of cholesterol in feed can regulate NPC expression, as well as promote the ovarian maturation and molting. The regulatory mechanism of NPC1in ovarian maturation, such as synthesis site in the hepatopancreas, cholesterol transport routes need to be elucidated in further researches. 

  1. Conclusion, adjust this section according to my previous comments.

  Response: Thanks for your good comments. We have adjust conclusion according to your previous comments.

This is the first study of NPC1 in crustaceans. The sequence results suggested that Mn-NPC1 is functionally conserved with other vertebrates and invertebrates. In contrast to insects, the Mn-NPC1 expression results showed that Mn-NPC1 plays a critical biological role in adults, but not in embryos or larvae. In M. nipponense, Mn-NPC1 was mainly dis-tributed in the hepatopancreas of adult females, and its expression was positively corre-lated with ovarian development. Furthermore, it played important biological roles in both ovarian maturation and molting in M. nipponense by regulating ecdysone. The results of this study expand our knowledge of NPC1 in crustaceans and provide further information regarding the regulatory mechanisms of ovarian maturation in M. nipponense. The regulatory mechanism of NPC1in ovarian maturation, such as synthesis site in the hepatopancreas, cholesterol transport routes need to be elucidated in further researches.

Round 2

Reviewer 3 Report (New Reviewer)

Comments and Suggestions for Authors

Thanks for your reply. I am pleased with this new version of your manuscript.

This manuscript is a resubmission of an earlier submission. The following is a list of the peer review reports and author responses from that submission.

Round 1

Reviewer 1 Report

Comments and Suggestions for Authors

General comments

In this paper, the authors investigated the cloning of a cDNA encoding NPC intracellular cholesterol transporters 1 (Mn-NPC1) from the oriental river prawn Macrobrachium nipponense. Mn-NPC1 was highly expressed in the hepatopancreas, moderately in the ovary and heart. Expression levels of Mn-NPC1 in the ovary positively correlated with vitellogenesis. Gene silencing of Mn-NPC1 reduced GSI and 20-hydroxyecdysone (20E) content in the ovary. Therefore, the authors suggest that Mn-NPC1 may control ovarian maturation. There are, however, some serious issues described below, and therefore this article at this form might not be acceptable for the publication in International Journal of Molecular Sciences.

Specific comments

1) Lines 59-71, the explanation of NPC1 is insufficient. Adding details such as NPC1 biological functions and expression tissues in invertebrates with specific research examples would help the reader understand NPC1 in invertebrates better.

2) The authors showed the gene expression of Mn-NPC1 only in females. It is not clear why it was not examined in males.

3) Despite no relation between 20E content in the ovary and molting, why the authors measured 20E in the ovary rather than in the hemolymph? Additionally, information on the ELISA kits used in this study was not obtained through my search. The web page or manufacture’s instruction should be provided.

Author Response

Dear Reviewer,

Thank you very much for your comments and suggestions. The valuable comments from you not only helped us with the improvement of our manuscript, but suggested some ideas for future studies.

Below you will find our responses to your comments:

  1. Lines 59-71, the explanation of NPC1 is insufficient. Adding details such as NPC1 biological functions and expression tissues in invertebrates with specific research examples would help the reader understand NPC1 in invertebrates better.

Response: Thanks for your valuable suggestion. We added some details about NPC1 biological functions and expression tissues in invertebrates with specific research examples to help the reader understand better. Please find in the in the revised manuscript.

NPC1 is conserved and widely distributed in both vertebrates and invertebrates [14]. In vertebrates, low-density lipoprotein particles are taken up by cells and sent to lysosomes for decomposition by acid lipase [15-16]. The cholesterol released is exported by transporters (NPC1 and NCP2) to cells via the lysosomes [17-18]. In vertebrates, NPC1 is highly expressed in the intestinal epithelial cells. An inherited autosomal disease results in mutations of NPC1, and causes significant accumulation of cholesterol and glycosphingolipids, [19]. The NPC1 protein has only been rarely reported in insects. Two vertebrate homologues (NPC1a and NPC1b) have been found in Drosophila spp. NPC1a is widely distributed in the tissues whereas NPC1b is only expressed in the midgut epithelial cells. NPC1a is responsible for intracellular sterol transport, while NPC1b is involved in dietary sterol absorption [20-21]. NPC1 has also been shown to be involved in spermatogenesis in Drosophila spp. Specifically, male sterility was completely rescued using transgenic techniques using a widely expressed promoter to drive NPC1 gene expression in the NPC1 mutant [22]. In Bombyx mori, the expression of BmNPC1 was greatest in the testis, malpighian tubules, hemocytes, and ovaries. Knockdown of BmNPC1 caused cholesterol to accumulate in the cells and also prevented molting in the larvae [23]. In Bemisia tabaci, BtNPC1 was widely expressed at all developmental stages and the highest level was found in the adult abdomen. BtNPC1 knockdown led to reduced survival and fecundity. Knockdown of BtNPC1 also affected the development and metamorphosis of B. tabaci nymphs [24]. However, NPC1 has not yet been studied in crustaceans..

  1. The authors showed the gene expression of Mn-NPC1 only in females. It is not clear why it was not examined in males.

 Response: As we wrote in the “Introduction”, the purpose of this study was mainly focused on the exploration the potential mechanisms of fast ovary development in M. nipponense. In previous research of our team, comparative hepatopancreatic proteomic analysis of female prawn during five ovarian maturation stages were performed to explore differentially expressed proteins and pathways regulating ovarian maturation. An NPC1 protein was found to be listed as the top up-regulated protein in vitellogenesis stage (ovary stage II to III) and top down-regulated protein in emptying stage (ovary stage IV to V), indicating its significant roles in regulating ovarian maturation. Therefore, this study was aimed to investigate the function of Mn-NPC1 in ovarian maturation. The function of Mn-NPC1 in male will be studied in further researches.

  1. Despite no relation between 20E content in the ovary and molting, why the authors measured 20E in the ovary rather than in the hemolymph? Additionally, information on the ELISA kits used in this study was not obtained through my search. The web page or manufacture’s instruction should be provided.

 Response: Thanks for your kind suggestions. In crustaceans, free ecdysteroids including 20-hydroxyecdysone and ecdysone have been proved to be present in both ovaries and hemolymph (Young et al., 1993a; Young et al., 1993b). There are also studies to prove that haemolymph ecdysteroid titres relate to changes in molting status but not to changes in oocyte development and the ovary are capable of de novo synthesis of the ecdysteroids (Styrishave et al., 2008). In this study, we mainly focused on the potential roles of Mn-NPC1 in ovarian maturation in M. nipponense.  Therefore, we measured the ecdysteroids content in the ovary. According to your kindly reminder, we find that we made mistakes in our presentation of ecdysteroids measurement. We used the Penaeus Ecdysterone ELISA Kit to measure the ecdysterone content. We revised the presentation of ecdysteroids in the manuscript for ecdysterone content. The manufacture’s instruction of the Kit is attached in revised package.

Young, N. J., Webster, S. G., & Rees, H. H. (1993) a. Ovarian and hemolymph ecdysteroid titers during vitellogenesis in Macrobrachium rosenbergii. General and comparative endocrinology, 90(2), 183-191.

Young, N. J., Webster, S. G., & Rees, H. H. (1993) b. Ecdysteroid profiles and vitellogenesis in Penaeus monodon (Crustacea: Decapoda). Invertebrate reproduction & development, 24(2), 107-117.

Styrishave, B., Lund, T., & Andersen, O. (2008). Ecdysteroids in female shore crabs Carcinus maenas during the moulting cycle and oocyte development. Journal of the Marine Biological Association of the United Kingdom, 88(3), 575-581.

Reviewer 2 Report

Comments and Suggestions for Authors

In this manuscript the authors aanalyzed cDNA sequence and amino acid sequence of NPC1 in M. nipponense .Tissue and developmental stage expression profiles and subcellular localizaion were included. Also a few fucntional experimnets were included. The English is really so poor in this manuscript so it was very difficult to review it . Such a poorly edited  manuscript must be fully edited into readable Englsih before being sent out and I recomend auhtors to edit Englsih and then resubmit it.

Figure 1 and 2 should be in Supplement or at least made readable.

Figure 3 Crustacea not Crustacean  is correct in this Figure

Figure 5 is too small to be able to see what is shown!

Are really the differences in Figure 7 A and B as also  8A and B significantly different? Especially data in Figure 7B and 8 B seem not so dramtiaclly different! From how many individual animals were each sample point made? In other words are these true biological replicates?

Comments on the Quality of English Language

The English is very poor and needs a complete editing by a professional English editor.

Author Response

Dear Reviewer,

Thank you very much for your comments and suggestions. The valuable comments from you not only helped us with the improvement of our manuscript, but suggested some ideas for future studies.

Below you will find our responses to your comments:

  1. Figure 1 and 2 should be in Supplement or at least made readable.

Response: Thanks for your valuable suggestion. We put Figure 1and 2 to supplementary figures.

  1. Figure 3 Crustacea not Crustacean  is correct in this Figure.

Response: Thanks for your kindly reminding. We corrected the word in the Figure 3.

  1. Figure 5 is too small to be able to see what is shown!

Response: We resubmitted all original high-definition vector images together with the revised manuscript.

  1. Are really the differences in Figure 7 A and B as also  8A and B significantly different? Especially data in Figure 7B and 8 B seem not so dramtiaclly different! From how many individual animals were each sample point made? In other words are these true biological replicates?

 Response: First of all, thanks for your kindly reminding. We re-examined Figures 7A and 7B and found that the titles of y-axis was wrong which should be percentage values for GSI and the cumulative proportion of ovary stage â…¢. We corrected this in new Figure 5 in the revised manuscript.

In the long-term RNAi experiment, 180 healthy female prawns (1.57±0.22g) in ovary stage â…  were selected and randomly divided into two groups: control group and experimental group. Each group contained 3 parallel lines (n =30) (water temperature 24±1 ℃). Both two groups were injected with ds-GFP and ds-NPC1 at a dose of 4 μg/g b.w, respectively and the injections were given every 5 days. Totally nine prawns (three biological replicates in each parallel line) were randomly sampled from each group at the 1st, 9th 17th and 25th day, and their ovary and body weight were measured immediately to calculate GSI. After that, the dissected ovaries were used for ecdysterone content determination. There were statistically significant differences in these parameters by two-tailed t-test.

  1. The English is very poor and needs a complete editing by a professional English editor.

Response: Thanks for your kindly reminding. We have corrected many orthographical errors, spelling errors and mistakes according to the comments and suggestions of reviewers and editor. The revised manuscript has been edited and proofread by an editing company in Ireland.

Round 2

Reviewer 2 Report

Comments and Suggestions for Authors

This manuscript has been revised but the original recommendation was reject and still this is the present recommendation to reject the manuscript since no functional studies gave been added.

Comments on the Quality of English Language

poor needs a complete editing